# Onchocerciasis and non-communicable diseases in the Bafut Health District, Cameroon: Knowledge, attitudes, and practices towards community-directed treatment with ivermectin

Irene U. Ajonina-Ekoti[1,2,3], Promise A. Aghaeze[1], Joan Ebanga[1], Theophilus A. Ekoti[2,4], Tiburce Gangue[5], Beri A. Gariba[6], Moses A. Mbanwi[7], Adolph A. Fozao[7], Carine K. Nfor[2], Mbunkah D. Achukwi[3], Marcelus U. Ajonina [2,8,9,10]*

**1** Department of Microbiology and Parasitology, The University of Bamenda, Bambili, Cameroon, **2** McGadi Education and Research Initiative, Buea, Cameroon, **3** TOZARD Laboratory, Bambili, Cameroon, **4** Department of Educational Psychology, The University of Bamenda, Bambili, Cameroon, **5** Department of Zoology, The University of Bamenda, Bambili, Cameroon, **6** Northwest Regional Delegation of Public Health, Bamenda, Cameroon, **7** District Health Service, Bafut, Cameroon, **8** Graduate School of Health and Biomedical Sciences, St Louis University Institute, Douala, Cameroon, **9** Department of Public Health, The University of Bamenda, Bambili, Cameroon, **10** School of Health Sciences, Charisma University, Billings, Montana, United States of America

* majonin@gmail.com

## Abstract

### Background

Onchocerciasis remains a public health problem in Cameroon despite years of community-directed treatment with ivermectin (CDTI). Its persistence increasingly overlaps with the rising burden of non-communicable diseases (NCDs). Community knowledge, attitudes, and practices (KAP) toward onchocerciasis and CDTI, as well as the coexistence of chronic NCDs, may influence the success of elimination programs.

### Methodology

A community-based cross-sectional study was conducted between April and June 2022 in Bafut Health District, Northwest Cameroon. Using a structured, interviewer-administered questionnaire, information was collected on sociodemographic characteristics, knowledge, attitudes, and practices related to onchocerciasis and CDTI, as well as symptoms and comorbidities associated with NCDs. Descriptive statistics were used to summarize responses, while the association between quantitative variables was determined using the Chi-square test and logistic regression analysis. Statistical significance was set at $p < 0.05$.

### Results

Of the 250 respondents (mean age $40.9 \pm 13.6$ years), 96.8% had heard of onchocerciasis, though only 46.0% correctly identified the filarial worm as the cause. Good

**Data availability statement:** All relevant data are within the manuscript and its Supporting information files. The structured questionnaire used for data collection is provided as Supporting information (S1 File). The de-identified dataset underlying the findings of this study is provided as Supporting information (S1 Data).

**Funding:** The author(s) received no specific funding for this work.

**Competing interests:** The authors have declared that no competing interests exist.

knowledge, a positive attitude, and good practices regarding onchocerciasis and CDTI were observed in 78.0%, 61.3%, and 90.4% of the respondents, respectively. Symptoms of onchocerciasis were reported by 188 participants (75.2%), with itching being the most common (50.4%). Of these, 46 (24.5%) also reported a diagnosis of NCD, while 84 (44.7%) reported a family history of NCDs. Arthritis (32.4%) was the most common self-reported NCD, and hypertension (14.0%) was the most frequent family history.

## Conclusions

This study revealed high levels of knowledge and good practices regarding onchocerciasis and CDTI; however, knowledge gaps and concerns about side effects continue to hinder the uptake of ivermectin. The coexistence of onchocerciasis with chronic NCDs highlights the need for integrated disease management and reinforced health education to support elimination goals.

### Author summary

Onchocerciasis, or river blindness, remains a significant public health concern in Cameroon despite decades of community-directed treatment with ivermectin (CDTI). Although CDTI has reduced transmission, the disease continues to affect communities where non-communicable diseases (NCDs) such as arthritis, hypertension, diabetes, stroke, and epilepsy are also on the rise. The coexistence of onchocerciasis and chronic conditions may complicate treatment decisions, affect adherence, and undermine elimination goals. This study was conducted among 250 residents of Bafut Health District, where onchocerciasis is endemic. Using a structured questionnaire, knowledge, attitudes, and practices (KAP) toward onchocerciasis and CDTI were assessed while documenting the occurrence of NCDs. Most respondents demonstrated good knowledge of the disease and reported taking ivermectin. Over 75.2% of participants reported symptoms suggestive of onchocerciasis, and nearly one in four of these also had an NCD diagnosis, while close to half reported a family history of NCDs. Arthritis was the most common condition reported, and hypertension was the most frequent family history. The findings highlight the need for integrated approaches that link onchocerciasis elimination with NCD prevention and management, supported by strengthened community health education.

## Introduction

Onchocerciasis, also known as river blindness, is a major neglected tropical disease (NTD) that continues to impose a substantial public health and socioeconomic burden. Globally, an estimated 21 million individuals are infected, with more than 99% of cases occurring in sub-Saharan Africa [1]. The disease is caused by the filarial

parasite *Onchocerca volvulus*, transmitted to humans through the bite of infected female blackflies (Simulium spp.), which breed along fast-flowing rivers and streams [2]. During a blood meal, the fly introduces infective third-stage larvae into the human host, where they develop into adult worms that reside in subcutaneous tissues. Female worms can release thousands of microfilariae daily, which are responsible for the dermatological and ocular manifestations of the disease [3]. According to the Global Burden of Disease Study, as of 2017, approximately 20.9 million people were infected, of whom 14.6 million had cutaneous disease and 1.15 million were visually impaired or blind [4].

Cameroon remains one of the most endemic countries, with onchocerciasis present in all ten regions. An estimated six million individuals are infected, and more than half of the population resides in high-risk rural areas along rivers where blackflies are abundant [5,6]. Clinical manifestations include severe itching, skin depigmentation and disfigurement, ocular lesions, and progressive loss of vision. The condition is the second leading cause of infectious blindness worldwide after trachoma. Beyond physical suffering, stigmatization and social exclusion are common, particularly among women and young people, who may face reduced marriage prospects and diminished social participation [7].

Despite decades of control efforts, including vector control and large-scale mass drug administration (MDA) with ivermectin through the community-directed treatment with ivermectin (CDTI) strategy, onchocerciasis persists in several endemic foci. Persistent transmission has been documented in multiple endemic settings, highlighting heterogeneity in elimination progress and the presence of transmission hotspots even after repeated rounds of ivermectin treatment [8] Projections from epidemiological models suggest that elimination by 2030 may not be achievable if MDA remains the sole intervention [9,10]. Several challenges hinder elimination goals: low adherence to ivermectin, treatment refusals due to adverse effects, limited program reach in conflict-affected areas, and interruptions caused by sociopolitical instability and environmental barriers [11,12]. Misconceptions and knowledge gaps about the cause and transmission of onchocerciasis continue to weaken community engagement in control efforts [11,12].

Coinfections and comorbidities also complicate efforts to eliminate the disease. In particular, ivermectin can trigger severe adverse events in individuals co-infected with *Loa loa* [13,14]. Additionally, the rising burden of non-communicable diseases (NCDs), including diabetes, hypertension, arthritis, epilepsy, stroke, and cancers, presents new challenges for endemic communities [4,15]. Certain chronic conditions share clinical features with onchocerciasis, including visual impairment and reduced mobility, which may complicate symptom recognition in endemic settings. In addition, an increasing body of evidence from Cameroon and other sub-Saharan African countries has demonstrated an association between *Onchocerca volvulus* infection and epilepsy, commonly referred to as onchocerciasis-associated epilepsy [16–19]. The coexistence of onchocerciasis with non-communicable diseases (NCDs) may hinder elimination efforts by reducing adherence to community-directed treatment with ivermectin (CDTI). Individuals with chronic conditions may avoid ivermectin due to fear of adverse effects or symptom overlap with existing illnesses, potentially lowering treatment coverage and delaying transmission interruption [11,20,21].

Community perceptions regarding onchocerciasis and its management remain critical for the success of elimination programs. Understanding how endemic populations perceive the disease, its treatment, and its overlap with NCDs is essential for designing sustainable, context-specific strategies. However, little research has systematically explored these issues, particularly in the Bafut Health District. Despite decades of mass drug administration, the overlap of onchocerciasis with rising NCDs in endemic communities remains poorly characterized. This study, therefore, aims to assess community knowledge, attitudes, and practices regarding onchocerciasis, while also examining its comorbidities with NCDs in Bafut, Cameroon.

## Methods

### Ethical considerations

The University of Bamenda Institutional Review Board (No. 2022/078H/UBa/IRB), the Northwest Regional Delegation of Public Health, and the District Medical Officer (DMO) of the Bafut Health District approved the study protocol. Written

informed consent was obtained from all respondents, either by signature or thumbprint, after a detailed explanation of the study's nature and objectives. Participation was entirely voluntary. Data collected was treated with strict confidentiality and stored on password-protected computers. All participants were handled in accordance with the principles of the Declaration of Helsinki on the ethical use of humans in biomedical research.

### Study setting and population

This was a community-based cross-sectional study conducted at the Bafut Health District (BHD) from April to June 2022. The BHD is located in the Mezam Division (6°10′0.000″N, 10°6′0.000″E), Northwest Region, about 20 km from Bamenda town along a 10-kilometre stretch of the "Ring Road" that trails along a ridge above the Menchum Valley [22]. The BHD is one of the twenty-one health districts in the northwest region [23], with fourteen health areas. According to the 2004 Rapid Epidemiological Assessment (REA) of onchocerciasis by the Bafut District Health Service, these health areas are classified as hypoendemic, meso-endemic, or hyperendemic [24]. The district has an estimated population of over 70,000 inhabitants, predominantly rural, with farming, fishing, hunting, trading, and small-scale businesses being the major sources of livelihood [25].

Bafut is the most powerful of the traditional kingdoms of the Grassfields, now divided into 26 wards. Administratively, the Bafut Sub-Division is one of the five sub-divisions in the Mezam Division, covering a land area of approximately 340 km$^2$. Geographically, it lies within the western grass fields ecological zone, which extends across the Northwest Region of Cameroon and neighboring highland savannah areas. The River Mezam and its tributaries drain the sub-division: the central collector and its tributaries form a relatively sparse, rectangular drainage network. The presence of ecological features such as forested landscapes with fast-flowing rivers creates suitable habitats for Onchocerca vectors, contributing significantly to the persistence of onchocerciasis. The area also experiences high rainfall, elevated temperatures, and intense human activity, all of which contribute to the proliferation of vectors and the persistence of disease.

The study population consisted of adult residents aged 18 years and above living in the Bafut Health District. Adults were selected because they are the primary decision-makers regarding participation in community-directed treatment with ivermectin (CDTI), are eligible to provide informed consent, and are more likely to reliably report prior exposure to ivermectin and diagnoses of non-communicable diseases. Only individuals who had resided in the community for at least 12 months were eligible to participate, to ensure adequate exposure to CDTI activities and local health education interventions.

### Sample size and sampling method

The minimum sample size was determined using Cochran's formula for estimating a single population proportion at a 95% confidence level and a 5% margin of error, as follows:

$$n = \frac{Z^2 pq}{d^2}$$

where $z^2$ = (1.96)2, p = previous knowledge of onchocerciasis and q = 1-p, $d^2$ = (0.05)$^2$. Based on a previous study conducted in the Centre Region of Cameroon, Domche et al. [26] reported a knowledge level of 91.6% regarding onchocerciasis and blackfly nuisance. Using this estimate, a minimum sample size of 160 participants was obtained. To account for potential non-response, an additional 10% was added, resulting in a final minimum sample size of 176 participants.

A multistage sampling approach was used. First, communities within the Bafut Health District were selected based on their endemicity for onchocerciasis. Within selected communities, households were systematically sampled. To avoid clustering and over-representation of responses within households, only one eligible adult respondent was selected per household. When more than one eligible adult was present, a simple random selection method (ballot method) was used

to identify the participant. If the selected individual declined participation or was unavailable, the household was revisited once before replacement was made.

## Data collection instrument

Data was collected using a pre-tested, structured, interviewer-administered questionnaire comprising predominantly closed-ended questions. The questionnaire was developed on previously validated knowledge, attitudes, and practices (KAP) studies on onchocerciasis [18,26–28] and adapted to include NCD-related components (S1 File). It was used to document information on sociodemographic characteristics of participants, knowledge of onchocerciasis and community-directed treatment with ivermectin (CDTI), attitudes and practices toward onchocerciasis and CDTI, as well as symptoms and non-communicable diseases.

The questionnaire consisted of 31 items. Eight items captured sociodemographic information, including gender, age, education level, marital status, religion, health area (endemicity), and duration of residence in the community. Nine items assessed knowledge of onchocerciasis and CDTI, including awareness, transmission, symptoms, prevention, ivermectin distribution, and management of side effects. Five items measured attitudes, covering perceptions of disease severity, ivermectin effectiveness, acceptance of annual treatment, and the role of traditional medicine. Four items addressed practices, focusing on ivermectin use, timing of last intake, side effects, and reasons for refusal. The final five items explored symptoms and NCDs, documenting self-reported onchocerciasis symptoms, prior diagnoses of major NCDs, and relevant family history.

Interviews were conducted face-to-face in English or, where necessary, in local languages. Data collection was carried out by trained research assistants who received prior instruction on standardized questionnaire administration, ethical considerations, and maintaining interviewer neutrality to ensure consistency and minimize bias. Community drug distributors were not involved in data collection in order to reduce potential response bias associated with their role in community-directed treatment with ivermectin.

## Variables

**Dependent variables.** The outcome variables for this study were knowledge of onchocerciasis and CDTI, attitudes toward the disease and its treatment, preventive practices, and reported onchocerciasis-NCD comorbidity. Knowledge was assessed using eight core questions covering awareness, transmission, symptoms, prevention, and ivermectin distribution. A respondent who had correct responses for at least five of the components was considered to have good knowledge, while those with correct responses to four or fewer were considered to have poor knowledge regarding onchocerciasis and CDTI.

Attitudes were assessed using five items: perception of onchocerciasis as a serious disease, belief in the effectiveness of ivermectin, willingness to encourage family or friends to take ivermectin, comfort with annual ivermectin intake, and reliance on traditional medicine. A response of "Yes" to items on disease seriousness, ivermectin effectiveness, willingness to encourage others, and comfort with annual intake was coded as positive, while a response of "No" or "I don't know" was coded as negative. For the traditional medicine item, "No" was coded as positive, while "Yes" and "I don't know" were coded as negative. A respondent with three or more positive responses was classified as having an overall positive attitude; otherwise, the respondent was classified as having a negative attitude.

Practices were assessed using six items, of which three were used for scoring. These included whether respondents had ever taken ivermectin, the timing of their most recent dose, and their willingness to take ivermectin anytime it was recommended. Ever taking ivermectin, taking it within the past year or a few months, and being willing to take it anytime recommended were coded as right practices, while alternative responses were coded as wrong practices. The experience of side effects and the type of symptoms reported were treated descriptively, without classifying them as right or wrong. A respondent with two or more right practices was classified as having good practice; otherwise, they were classified as

having poor practice. The cut-off points used for classifying good knowledge, positive attitude, and good practices were adapted from similar KAP studies in Cameroon and Nigeria [26,27].

The comorbidity domain captured self-reported symptoms of onchocerciasis (itching, nodules, blurred vision, skin changes) alongside prior diagnosis or family history of selected NCDs, including hypertension, diabetes, stroke, arthritis, and epilepsy.

**Independent variables.** Explanatory variables were limited to the socio-demographic factors including; gender (male, female), age group (<30, 31–50, > 50 years), marital status (single, married), level of education (none, primary, secondary, tertiary), occupation (student, farmer, employed, business), religion (Christian, Muslim, Atheist), duration of residence in the area (<2 years, 3–10 years, > 10 years), and health area endemicity (hypo-, meso-, hyper-endemic).

### Data analysis

All completed questionnaires were checked for completeness, coded, and entered into Microsoft Excel before being exported to SPSS Statistics version 27.0 (IBM Corp., Armonk, NY, USA) for analysis. Descriptive statistics such as frequencies, percentages, means, and standard deviations were used to summarize socio-demographic characteristics, knowledge, attitudes, practices, and reported symptoms and NCDs.

Associations between categorical variables, including socio-demographic factors and knowledge, attitude, and practices regarding onchocerciasis and CDTI, were examined using Pearson's $\chi^2$ test. To determine factors associated with good knowledge, positive attitude, and good practices towards onchocerciasis and CDTI, logistic regression analysis was performed. Variables with a p-value ≤0.10 in bivariate analysis were entered into the multivariable logistic regression model. Adjusted Odds Ratios (AOR) with their 95% confidence intervals (CI) were calculated to assess the strength of association. Statistical significance was set at $p < 0.05$.

## Results

A total of 250 participants with a mean age (±SD) of 40.96 ± 13.59 years were enrolled in the study. The majority were female (54.0%), within the 31–50-year age group (50.4%), married (65.2%), had a secondary level of education (64.0%), of the Christian religion (88.0%), with farming as the predominant occupation (49.6%) (Table 1). Most participants reported having lived in the area for more than 10 years (69.6%), with a higher proportion residing in meso-endemic areas (46.4%).

### Knowledge regarding onchocerciasis and CDTI

Almost all participants (96.8%) acknowledged having heard about onchocerciasis (Table 2), however, only 46.0% correctly identified the filarial worm as the cause of the disease, while 35.6% attributed it to blackflies, and 12.4% to mosquitoes; a small proportion linked it to poor hygiene (1.6%). Regarding transmission, the majority (58.8%) correctly identified that blackfly bites are responsible, but 16.0% incorrectly attributed the cause to mosquito bites. On clinical presentation, itching (66.0%), skin changes (32.0%), and nodules (37.6%) were commonly recognised, though only 8.4% mentioned oedema. When asked about the possibility of person-to-person transmission of onchocerciasis, 38.3% of respondents believed it was possible, whereas 59.2% correctly indicated that it was not, and 2.5% reported being uncertain.

Awareness of prevention was generally high, with 85.6% of respondents acknowledging that onchocerciasis can be prevented, while 6.0% considered it not preventable and 8.4% were uncertain. The primary source of information on ivermectin distribution was community drug distributors (CDDs) (46.4%), while neighbours accounted for the lowest proportion (2.4%) (Fig 1). Regarding preventive measures, 50.6% cited ivermectin, 27.5% mentioned the use of protective clothing, 18.8% referred to avoiding bathing in rivers, 23.5% indicated environmental sanitation, and 15.4% suggested using bed nets. Awareness of annual ivermectin distribution was widespread (89.6%), and a majority (83.4%) were also aware that any side effects of ivermectin are managed free of charge.

**Table 1. Sociodemographic characteristics of the participants (N = 250).**

| Variable | Category | Number | Percent |
|---|---|---|---|
| Gender | Male | 115 | 46.0 |
| | Female | 135 | 54.0 |
| Age group | <30 | 60 | 24.0 |
| | 31-50 | 126 | 50.4 |
| | >50 | 64 | 25.6 |
| Mean age (mean ± SD) | 40.96 ± 13.59 | | |
| Level of education | None | 19 | 7.6 |
| | Primary | 71 | 28.4 |
| | Secondary | 87 | 34.8 |
| | Tertiary | 73 | 29.2 |
| Marital status | Single | 87 | 34.8 |
| | Married | 163 | 65.2 |
| Occupation | Employed | 46 | 18.4 |
| | Farmer | 124 | 49.6 |
| | Business | 37 | 14.8 |
| | Student | 43 | 17.2 |
| Religion | Christian | 220 | 88.0 |
| | Muslim | 18 | 7.2 |
| | Atheist | 12 | 4.8 |
| Endemicity | Hypo | 84 | 33.6 |
| | Meso | 116 | 46.4 |
| | Hyper | 50 | 20.0 |
| Duration lived in the area | <2 years | 25 | 10.0 |
| | 3-10 years | 51 | 20.4 |
| | >10 years | 174 | 69.6 |

Overall, the majority, 195 (78.0%) of respondents had a good level of knowledge, while 55 (22.0%) had a poor level of knowledge on onchocerciasis and CDTI. Analysis revealed that age, education, and endemicity were significantly associated with knowledge of onchocerciasis and CDTI (p < 0.05). Multivariate analysis showed that respondents aged over 50 years had the highest level of good knowledge (95.3%) and were six times more likely to demonstrate good knowledge compared to those under 30 years (aOR = 5.62, 95% CI: 1.51–20.88, p = 0.010). In contrast, participants aged 31–50 were less likely to report good knowledge (aOR = 0.42, 95% CI: 0.21–0.86, p = 0.017) (Table 3). Moreover, our results further showed that educational attainment is also positively associated with knowledge. Participants with tertiary education had the highest knowledge levels (86.3%) and were almost six times more likely to demonstrate good knowledge than those without formal education (aOR = 5.67, 95% CI: 1.85–17.39, p = 0.002). Similarly, respondents with primary education were more than three times as likely to report good knowledge (aOR = 3.66, 95% CI: 1.25–10.72, p = 0.018).

Endemicity level of residence was strongly associated with knowledge, with respondents in hypo-endemic areas being over three times more likely to report good knowledge compared to those in hyper-endemic areas (OR = 3.12, 95% CI: 1.31–7.45, p = 0.010). Those in meso-endemic areas also had higher knowledge (75.0%) compared to hyper-endemic regions, although this difference was not statistically significant (p < 0.05).

## Attitudes toward onchocerciasis and CDTI

Attitude of respondents towards onchocerciasis and CDTI was assessed using five questions (Table 4). Of all the respondents, less than half (39.2%) considered onchocerciasis a severe disease, while 44.0% did not, and 16.8% were uncertain

**Table 2. Knowledge of onchocerciasis and CDTI (N = 250).**

| Indicative questions | Response categories | n (%) |
|---|---|---|
| Have you ever heard about onchocerciasis? | Yes | 242 (96.8) |
| | No | 8 (3.2) |
| What causes onchocerciasis?* | Filarial worm | 115 (46.0) |
| | Blackfly | 89 (35.6) |
| | Mosquito | 31 (12.4) |
| | Poor personal hygiene | 4 (1.6) |
| | I don't know | 11 (4.4) |
| How are people infected with the disease?* | Blackfly bite | 147 (58.8) |
| | Contact with infected persons | 21 (8.4) |
| | Mosquito bite | 40 (16.0) |
| | Filarial worm | 4 (1.6) |
| | I don't know | 38 (15.2) |
| What are the signs and symptoms of onchocerciasis? * | itching | 165 (66.0) |
| | edema | 21 (8.4) |
| | skin change | 80 (32.0) |
| | Nodules in any part of the body | 94 (37.6) |
| | I don't know | 23 (9.2) |
| Can onchocerciasis be transmitted from person to person? | Yes | 92 (38.3) |
| | No | 148 (59.2) |
| | I don't know | 10 (2.5) |
| Is onchocerciasis preventable? | Yes | 214 (85.6) |
| | No | 15 (6.0) |
| | I don't know | 21 (8.4) |
| If yes, how can onchocerciasis be prevented? * | Wearing protective clothing | 68 (27.5) |
| | Avoiding bathing in rivers | 47 (18.8) |
| | Drugs (Ivermectin) | 130 (50.6) |
| | Environmental sanitation | 58 (23.5) |
| | Bednets | 38 (15.4) |
| | I don't know | 15 (6.0) |
| Are you aware of the annual ivermectin distribution? | Yes | 224 (89.6) |
| | No | 26 (10.4) |
| Do you know that the side effects of ivermectin are treated free of charge? | Yes | 186 (83.4) |
| | No | 37 (16.4) |

*Percentages may exceed 100% due to multiple responses.

(Table 4). Similarly, 43.0% of respondents perceived ivermectin as effective in preventing onchocerciasis, while 32.2% disagreed and 24.8% were unsure.

It was further observed that most respondents (88.3%) reported being comfortable with taking ivermectin annually, and over half (56.4%) indicated they would encourage family or friends to participate in CDTI. However, perceptions of the seriousness of onchocerciasis and the effectiveness of ivermectin were more divided, with fewer than half recognising the disease as severe (39.2%) or affirming ivermectin's preventive value (43.0%). Notably, nearly one-third (32.4%) believed traditional medicine could cure onchocerciasis, reflecting persistent misconceptions.

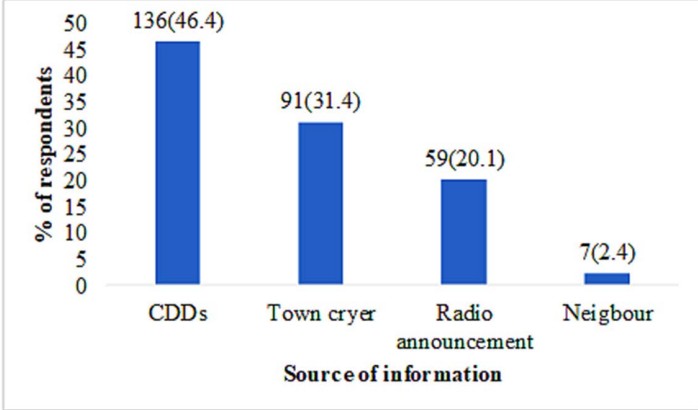

**Fig 1. Source of information on ivermectin distribution among respondents.**

**Table 3. Association between knowledge of onchocerciasis/CDTI and sociodemographic characteristics.**

| Variable | Category | N | Good knowledge [n (%)] | Unadjusted p-value | OR | 95%CI | Adjusted p-value |
|---|---|---|---|---|---|---|---|
| Gender | Male | 115 | 90 (78.3) | 0.817 | NA | | |
| | Female | 135 | 104 (77.0) | | | | |
| Age group | <30 | 60 | 47 (78.3) | <0.001 | 1 | | |
| | 31-50 | 126 | 76 (60.3) | | 0.42 | 0.21 – 0.86 | 0.017* |
| | >50 | 64 | 61 (95.3) | | 5.62 | 1.51 – 20.88 | 0.010* |
| Level of education | None | 19 | 10 (52.6) | 0.011* | 1 | | |
| | Primary | 71 | 57 (74.6) | | 3.66 | 1.25 – 10.72 | 0.018* |
| | Secondary | 87 | 61 (70.1) | | 2.11 | 0.77 – 5.80 | 0.147 |
| | Tertiary | 73 | 63 (86.3) | | 5.67 | 1.85 – 17.39 | 0.002 |
| Marital status | Single | 87 | 70 (80.5) | 0.428 | NA | | |
| | Married | 163 | 124 (76.1) | | | | |
| Occupation | Student | 43 | 36 (83.7) | 0.068 | 1 | | |
| | Employed | 46 | 37 (80.4) | | 0.8 | 0.27–2.37 | 0.686 |
| | Farmer | 124 | 88 (71.0) | | 0.48 | 0.19–1.23 | 0.127 |
| | Business | 37 | 33 (89.2) | | 1.61 | 0.47–5.53 | 0.446 |
| Religion | Christian | 220 | 171 (77.7) | 0.143 | NA | | |
| | Muslim | 18 | 16 (88.9) | | | | |
| | Atheist | 12 | 7 (58.3) | | | | |
| Endemicity | Hypo | 84 | 73 (86.9) | 0.026* | 3.12 | 1.31 – 7.45 | 0.010* |
| | Meso | 116 | 87 (75.0) | | 1.41 | 0.68 – 2.92 | 0.353 |
| | Hyper | 50 | 34 (68.0) | | 1 | | |
| Duration lived in the area | <2 years | 25 | 20 (80.0) | 0.379 | NA | | |
| | 3-10 years | 51 | 43 (84.3) | | | | |
| | >10 years | 174 | 131 (75.3) | | | | |

It was generally observed that 153 participants (61.3%) demonstrated a positive attitude, while 97 (38.7%) exhibited a negative attitude toward onchocerciasis and CDTI. Positive attitude toward onchocerciasis and CDTI was significantly associated with gender, education, religion, and endemicity (p<0.05) (Table 5). Female respondents were 1.3 times

**Table 4. Attitude of the respondent towards onchocerciasis and CDTI (N = 250).**

| Indicative questions | Response categories | n (%) |
|---|---|---|
| Do you think onchocerciasis is a serious disease in your community? | Yes | 98 (39.2) |
| | No | 110 (44.0) |
| | I don't know | 42 (16.8) |
| Do you think Ivermectin is effective in preventing onchocerciasis | Yes | 108 (43.0) |
| | No | 81 (32.2) |
| | I don't know | 61 (24.8) |
| Would you encourage family/friends to take ivermectin during CDTI? | Yes | 141 (56.4) |
| | No | 109 (43.6) |
| Do you think traditional medicine can cure onchocerciasis? | Yes | 81 (32.4) |
| | No | 109 (43.5) |
| | I don't know | 60 (24.1) |
| Are you comfortable taking ivermectin every year? | Yes | 219 (88.3) |
| | No | 31 (11.7) |

**Table 5. Association between attitude towards onchocerciasis/CDTI and sociodemographic characteristics.**

| Variable | Category | N | Positive attitude n (%) | Unadjusted p-value | OR | 95% CI | Adjusted p-value |
|---|---|---|---|---|---|---|---|
| Gender | Male | 115 | 82 (71.3) | 0.042* | 1 | | |
| | Female | 135 | 104 (77.0) | | 1.35 | 1.02–2.45 | 0.041* |
| Age group | <30 | 60 | 44 (73.3) | 0.118 | NA | | |
| | 31–50 | 126 | 91 (72.2) | | | | |
| | >50 | 64 | 48 (75.0) | | | | |
| Education level | None | 19 | 10 (52.6) | <0.001* | 1 | | |
| | Primary | 71 | 48 (67.6) | | 1.45 | 0.71–3.13 | 0.291 |
| | Secondary | 87 | 61 (70.1) | | 1.61 | 0.82–3.52 | 0.184 |
| | Tertiary | 73 | 63 (86.3) | | 3.12 | 1.45–6.74 | 0.003* |
| Marital status | Single | 87 | 67 (77.0) | 0.314 | NA | | |
| | Married | 163 | 113 (69.3) | | | | |
| Occupation | Student | 43 | 31 (72.1) | 0.229 | NA | | |
| | Employed | 46 | 35 (76.1) | | | | |
| | Farmer | 124 | 86 (69.4) | | | | |
| | Business | 37 | 29 (78.4) | | | | |
| Religion | Christian | 220 | 159 (72.3) | 0.037* | 1 | | |
| | Muslim | 18 | 16 (88.9) | | 2.45 | 1.02–7.88 | 0.037* |
| | Atheist/Other | 12 | 7 (58.3) | | 0.81 | 0.32–2.15 | 0.665 |
| Endemicity | Hypo | 84 | 61 (72.6) | 0.026* | 1 | | |
| | Meso | 116 | 84 (72.4) | | 1.08 | 0.65–1.86 | 0.552 |
| | Hyper | 50 | 42 (84.0) | | 1.92 | 1.12–4.35 | 0.029* |
| Duration lived in the area | <2 years | 25 | 19 (76.0) | 0.221 | NA | | |
| | 3–10 years | 51 | 39 (76.5) | | | | |
| | >10 years | 174 | 126 (72.4) | | | | |

more likely to demonstrate a positive attitude than males (aOR = 1.35, 95% CI = 1.02–2.45, p = 0.041). Participants with tertiary education were over three times more likely to have a positive attitude than those without formal education (aOR = 3.12, 95% CI = 1.45–6.74, p = 0.003). Respondents of the Muslim faith were also more likely to show positive attitudes than Christians (aOR = 2.45, 95% CI = 1.02–7.88, p = 0.037). In addition, residing in hyperendemic areas was significantly associated with a higher odds of a positive attitude compared to hypoendemic areas (aOR = 1.92, 95% CI = 1.12–3.35, p = 0.029). Other factors, including age, marital status, occupation, and duration of residence, were not significantly associated with attitude.

## Practices related to onchocerciasis prevention

In this study, the majority of respondents (93.2%) reported ever taking ivermectin, while 6.8% had never taken the drug (Table 6). Regarding the timing of uptake, 46.4% had last taken ivermectin within the previous year, 34.8% within a few months, while 10.0% had taken it two years prior. Among those who had ever taken the drug, 74.6% reported experiencing some form of side effect after treatment, while 25.4% did not. The most commonly reported side effects were body or eye itching (45.3%), joint pains (25.9%), body swelling (25.3%), and fever (16.5%). Of the 22 respondents who refused to take ivermectin, the main reasons cited were fear of side effects (68.4%), no longer having symptoms of onchocerciasis (18.2%), and a perception that treatment was unnecessary (13.6%). In addition, most respondents (87.6%) indicated they would be willing to take ivermectin anytime it is recommended, while 12.4% expressed unwillingness.

The majority of respondents [226(90.4%)] demonstrated good practices, while 24(9.6%) showed poor practices toward onchocerciasis prevention. Binary logistic regression revealed that age, endemicity, and community residence duration were significantly associated with practice level (Table 7). Respondents aged above 50 years were more likely to demonstrate good practices than those younger than 30 (aOR = 3.13, 95% CI: 1.04–9.39, p = 0.041). Living in hypoendemic areas was associated with better practices than residing in hyperendemic areas (aOR = 0.26, 95% CI: 0.07–0.91, p = 0.034). In addition, individuals who had lived in the community for more than ten years were more likely to report good practices than those who had resided for less than two years (aOR = 2.18, 95% CI: 0.69–6.85, p = 0.041). Other factors,

**Table 6. Practices regarding onchocerciasis prevention.**

| Indicative questions | Response categories | N | n (%) |
|---|---|---|---|
| Have you ever taken Mectizan (Ivermectin)? | Yes | 250 | 233 (93.2) |
| | No | | 22 (6.8) |
| When was the last time you took ivermectin? | A few months ago | 228 | 87 (34.8) |
| | Last year | | 116 (46.4) |
| | The last two years | | 25 (10.0) |
| Did you experience any side effects after taking ivermectin? | Yes | 228 | 170 (74.6) |
| | No | | 58 (25.4) |
| If yes, which side effects did you experience? | Body/eye itching | 170 | 77 (45.3) |
| | Body swelling | | 43 (25.3) |
| | Joint pains | | 44 (25.9) |
| | Fever | | 28 (16.5) |
| If you ever refused ivermectin, what was your reason? | Fear of side effects | 22 | 15 (68.4) |
| | No longer have symptoms of onchocerciasis. | | 4 (18.2) |
| | Did not see the need | | 3 (13.6) |
| Will you be willing to take ivermectin anytime recommended? | Yes | 250 | 219 (87.6) |
| | No | | 31 (12.4) |

**Table 7. Practices regarding onchocerciasis control (N = 250).**

| Variable | Category | N | Good practice n (%) | Unadjusted p-value | OR | 95% CI | Adjusted p-value |
|---|---|---|---|---|---|---|---|
| Gender | Male | 115 | 103 (89.6) | 0.644 | NA | | |
| | Female | 135 | 123 (91.1) | | | | |
| Age group | <30 | 60 | 52 (86.7) | 0.032 | 1 | | |
| | 31–50 | 126 | 113 (89.7) | | 1.32 | 0.51–3.41 | 0.572 |
| | >50 | 64 | 61 (95.3) | | 3.13 | 1.04–9.39 | 0.041 |
| Education level | None | 19 | 17 (89.5) | 0.217 | NA | | |
| | Primary | 71 | 64 (90.1) | | | | |
| | Secondary | 87 | 79 (90.8) | | | | |
| | Tertiary | 73 | 66 (90.4) | | | | |
| Marital status | Single | 87 | 78 (89.7) | 0.314 | NA | | |
| | Married | 163 | 148 (90.8) | | | | |
| Occupation | Student | 43 | 38 (88.4) | 0.229 | NA | | |
| | Employed | 46 | 41 (89.1) | | | | |
| | Farmer | 124 | 112 (90.3) | | | | |
| | Business | 37 | 35 (94.6) | | | | |
| Religion | Christian | 220 | 198 (90.0) | 0.154 | NA | | |
| | Muslim | 18 | 17 (94.4) | | | | |
| | Atheist/Other | 12 | 11 (91.7) | | | | |
| Endemicity | Hypo | 84 | 80 (95.2) | 0.027 | 1 | | |
| | Meso | 116 | 104 (89.7) | | 0.45 | 0.14–1.43 | 0.178 |
| | Hyper | 50 | 42 (84.0) | | 0.26 | 0.07–0.91 | 0.034 |
| Duration lived in the area | <2 years | 25 | 21 (84.0) | 0.043 | 1 | | |
| | 3–10 years | 51 | 45 (88.2) | | 1.43 | 0.38–5.35 | 0.596 |
| | >10 years | 174 | 160 (92.0) | | 2.18 | 0.69–6.85 | 0.041 |

OR=odds ratio; CI=confidence interval; 1=reference group; *Significant p values, NA=not applicable.

including gender, education level, marital status, occupation, and religion, were not significantly associated with practice level (p > 0.05).

## Onchocerciasis symptoms and non-communicable diseases

Symptoms of onchocerciasis were reported by 188 respondents (75.2%), while 62 (24.8%) indicated no symptoms (Table 8). The most frequently reported symptoms were itching of the skin or eyes (50.4%), followed by nodules (17.2%), blurred vision (16.8%), and skin changes (2.0%). Symptoms were experienced most often in the morning (25.2%) and afternoon (24.4%), with slightly fewer respondents reporting them at night (22.0%). With regard to non-communicable diseases (NCDs), 81 respondents (32.4%) reported having been diagnosed, while 169 (67.6%) reported no diagnosis. Among those diagnosed, arthritis was the most common (61.7%), followed by hypertension (21.0%), stroke (7.4%), epilepsy (4.9%), and diabetes (4.9%). A family history of NCDs was also common, with 150 respondents (60.0%) indicating at least one case in their family. The most frequently reported conditions were arthritis (18.8%), hypertension (14.0%), diabetes (11.2%), epilepsy (11.2%), and stroke (4.8%).

It was observed that, of the 188 respondents with onchocerciasis symptoms, 46 (24.5%) reported having an NCD diagnosis, while 84 (44.7%) indicated a family history of NCDs (Fig 2).

**Table 8. Onchocerciasis symptoms and non-communicable diseases.**

| Indicative questions | Response categories | N | n (%) |
|---|---|---|---|
| Do you have or sometimes present with symptoms of Onchocerciasis? | Yes | 250 | 188 (75.2) |
| | No | | 62 (24.8) |
| Which of the symptoms do you experience? | itching skin/eye | 188 | 126 (50.4) |
| | Nodules | | 43 (17.2) |
| | Blurred vision | | 42 (16.8) |
| | Skin changes | | 5 (2.0) |
| What time of the day do you mostly experience symptoms? | Morning | 188 | 66 (25.2) |
| | Afternoon | | 62 (24.4) |
| | Night | | 60 (22.0) |
| Have you ever been diagnosed with any non-communicable disease? | Yes | 250 | 81 (32.4) |
| | No | | 169 (67.6) |
| If yes, which non-communicable diseases have you been diagnosed with? | Stroke | 81 | 6 (7.4) |
| | Diabetes | | 4 (4.9) |
| | Hypertension | | 17 (21.0) |
| | Arthritis | | 50 (61.7) |
| | Epilepsy | | 4 (4.9) |
| Do you have a family history of any non-communicable disease? | Yes | 250 | 150 (60.0) |
| | No | | 100 (40.0) |
| If yes, which non-communicable diseases have been recorded in your family? | Stroke | 150 | 12 (4.8) |
| | Diabetes | | 28 (11.2) |
| | Hypertension | | 35 (14.0) |
| | Arthritis | | 47 (18.8) |
| | Epilepsy | | 28 (11.2) |

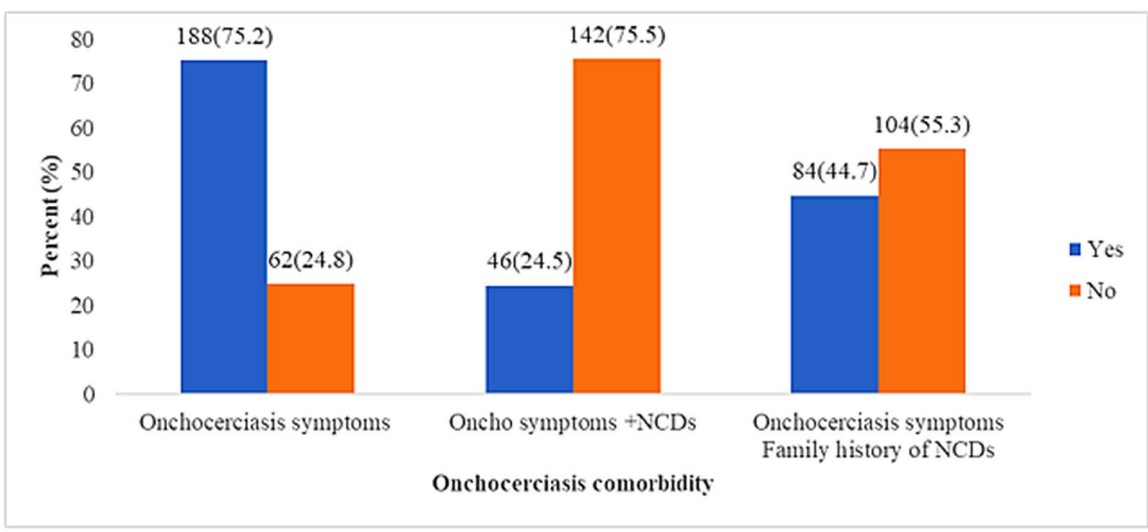

**Fig 2. Distribution of onchocerciasis symptoms, NCDs, and family history of NCDs.**

## Discussion

Despite decades of community-directed treatment with ivermectin (CDTI), onchocerciasis elimination in Cameroon remains threatened on an unprecedented scale by persistent transmission in endemic foci, suboptimal treatment coverage, reports of persistent transmission following multiple ivermectin treatment rounds, co-endemicity with other filarial infections, and the growing burden of non-communicable diseases (NCDs) [8,29]. Onchocerciasis remains endemic across all ten regions of Cameroon, affecting millions of people. Documented barriers to elimination include gaps in community knowledge, misconceptions regarding disease transmission, weaknesses in health system capacity, and disruptions related to sociopolitical instability, all of which continue to undermine the effectiveness of community-directed treatment with ivermectin (CDTI) [11,29,30]. In this context, the increasing prevalence of non-communicable diseases (NCDs) in endemic communities introduces additional complexity to efforts aimed at elimination [21,30].

Comorbid chronic conditions such as hypertension, arthritis, and epilepsy may contribute to apprehension toward ivermectin intake, particularly where symptoms of these conditions overlap with commonly reported post-treatment reactions such as joint pain, itching, or visual disturbances. Studies from Cameroon and other endemic settings have shown that fear of adverse effects and prior negative treatment experiences are important drivers of delayed uptake or non-compliance with ivermectin [20,29,31]. From a case management perspective, individuals with pre-existing illnesses who report post-treatment symptoms are more likely to require additional clinical assessment, reassurance, referral, or follow-up, thereby increasing the workload at peripheral health facilities. Programmatic evaluations of long-standing CDTI programs have documented challenges faced by health workers and community drug distributors in distinguishing ivermectin-related reactions from symptoms of underlying chronic conditions, which complicates counseling, pharmacovigilance, and community trust in treatment delivery [20,29,31]. In rural and resource-limited settings, these additional demands may constrain service capacity, reduce effective treatment coverage, and ultimately slow progress toward elimination, particularly where CDTI relies heavily on volunteer-based delivery systems [11,21,29].

Consistent with these programmatic challenges, the present study found that although awareness of onchocerciasis was high (96.8%), fewer than half of respondents (46.0%) correctly identified the filarial worm as the causative agent. Misattribution of the disease to blackflies, mosquitoes, or poor hygiene was a common misconception. Similar misconceptions have been reported in other endemic areas, where the role of blackflies as vectors is frequently conflated with the cause of infection [11]. Such misunderstandings may undermine confidence in elimination programs, particularly when combined with beliefs in person-to-person transmission or concerns about ivermectin-related side effects. Comparable concerns have been described in long-standing CDTI settings in Cameroon and have been associated with reduced adherence [29]. These findings underscore the importance of sustained community education that clearly explains the biological basis of the disease and reinforces messaging on ivermectin safety.

Knowledge of onchocerciasis and CDTI varied across socio-demographic groups. Older respondents and those with higher education levels were more likely to demonstrate a good understanding of onchocerciasis and CDTI, consistent with findings from the West Region of Cameroon and other endemic settings [32]. Respondents residing in hypoendemic areas also showed higher levels of knowledge than those in hyperendemic zones, a pattern that may reflect differences in program exposure, treatment coverage, and repeated health education activities rather than transmission intensity alone [28]. This difference may also reflect variations in programmatic exposure, including differences in treatment frequency, coverage, and the intensity of community sensitization activities, rather than the endemicity level alone. These disparities highlight the need for targeted strategies that prioritize younger individuals, those with limited formal education, and residents of high-transmission areas. Community drug distributors (CDDs), identified in this study as the primary source of information, represent an important channel for addressing these gaps; however, their effectiveness is greatest when their activities are well-coordinated with those of formal healthcare workers, particularly for counseling, referral, and management of treatment-related concerns. Strengthening collaboration between CDDs and health services may improve message consistency, community trust, and sustained participation in CDTI.

The present study revealed mixed community attitudes toward onchocerciasis and CDTI. Fewer than half of respondents perceived onchocerciasis as a severe disease (39.2%) or considered ivermectin effective in preventing it (43.0%). Such patterns have been reported in Cameroon and other endemic settings, where low perceived disease severity and doubts regarding ivermectin's effectiveness have been associated with reduced participation in mass drug administration programs [11,33]. Misconceptions regarding treatment options were also evident, with nearly one-third of respondents (32.4%) believing that traditional medicine could cure onchocerciasis, a perception that was similarly reported in studies from Nigeria and Uganda, where reliance on traditional remedies reduced uptake of ivermectin [34,35]. Despite these perceptions, most respondents reported comfort with taking ivermectin intake (88.3%), and willingness to encourage family and community members to participate in CDTI (56.4%), suggesting opportunities for strengthening behavior-change communication.

This study finds a significant association between positive attitudes and female gender, tertiary education, Muslim faith, and residence in hyperendemic areas. Similar observations were reported elsewhere, where education and other individual characteristics were found to shape long-term compliance with ivermectin distribution programs, with differences noted across gender and ethnic groups [36]. This association does not imply higher infection rates among women but may reflect gender-related differences in health-seeking behavior, caregiving roles, and engagement with community health activities, which can influence perceptions of preventive interventions. In Cameroon, non-compliance has also been linked to prior experiences with side effects, ethnicity, and the number of years an individual has lived in the community [20]. Beyond individual characteristics, community perceptions of how CDTI programs were organized and the commitment of community drug distributors (CDDs) were shown to be more strongly associated with adherence than disease beliefs themselves [29]. These findings suggest that improving adherence requires correcting misconceptions and strengthening the implementation system and procedures of the CDTI.

A higher proportion of respondents reported prior ivermectin use, and most expressed willingness to retake the drug if recommended. However, many also reported side effects, most commonly itching, joint pain, swelling, or fever, with fear of adverse effects emerging as a leading reason for treatment refusal. Similar findings have been reported in Ghana and Cameroon, where adverse reactions and prior negative experiences were identified as key predictors of non-compliance [20,31]. Evidence from other endemic settings further indicates that sustained uptake depends on effective pharmacovigilance, transparent dosing procedures, and active community engagement to address concerns about drug safety [29,36,37].

In our study, 75.2% of respondents reported symptoms consistent with onchocerciasis, most commonly itching of the skin or eyes (50.4%), nodules (17.2%), and blurred vision (16.8%). This pattern aligns with known morbidity profiles of onchocerciasis, where skin manifestations and ocular involvement are among the most frequent clinical expressions [38]. Modelling studies suggest that while acute symptoms, such as severe itching, tend to decline more rapidly with long-term ivermectin distribution, chronic conditions like skin changes and ocular damage persist due to cumulative parasite exposure [14]. The observation that many symptoms occurred in the morning and afternoon is consistent with the biting activity of *Simulium blackflies*, which are most active during daylight hours along fast-flowing river basins [4]. However, itching in onchocerciasis endemic settings is multifactorial. In addition to irritation following vector bites, chronic pruritus is a recognized manifestation of microfilarial migration and host inflammatory responses in the skin, which may occur independently of recent blackfly exposure. These overlapping mechanisms may complicate symptom interpretation for affected individuals and health workers, particularly in highly endemic areas and in populations with long-standing infection [13,37].

Non-communicable diseases were also frequently reported, with 32.40% of respondents indicating a diagnosis and 60.00% citing a family history. Arthritis was the most common NCD in this cohort, followed by hypertension, stroke, epilepsy, and diabetes, respectively. The comorbidity of onchocercal symptoms with chronic conditions is noteworthy, with nearly one-quarter (24.5%) of symptomatic individuals also reporting an NCD diagnosis, and almost half (44.70%)

citing a family history of NCD. While the direct causal links between onchocerciasis and most NCDs cannot be inferred from this cross-sectional study, there is substantial epidemiological evidence supporting an association between *Onchocerca volvulus* infection and epilepsy, commonly referred to as onchocerciasis-associated epilepsy (OAE), in endemic settings [15–18,38]. Apart from OAE, this study does not establish causal links between onchocerciasis and other non-communicable conditions, and the observed coexistence likely reflects shared geographic, environmental, and health system determinants rather than direct causation. The increasing recognition of the neurological consequences of chronic infection, along with evidence of sustained immune activation and inflammation associated with filarial disease, suggests potential interactions with pathways relevant to non-communicable conditions. These findings are consistent with the growing understanding that neglected tropical diseases and NCDs often overlap within the same populations, underscoring the need for integrated approaches that address both infection control and chronic disease management in endemic settings [21,30,39].

Given the comorbidity of onchocerciasis with chronic conditions such as arthritis and hypertension in this study, programmatic integration is needed. Linking CDTI platforms with community-based NCD screening and management could address dual disease burdens, improve treatment adherence, and strengthen primary health care. Such integrated approaches would enhance efficiency, reduce stigma, and advance both NTD elimination and NCD control objectives in Cameroon.

## Conclusion

This study highlights that despite widespread awareness of onchocerciasis in the Bafut Health District, persistent misconceptions about disease causation, transmission, and the safety of ivermectin continue to undermine elimination efforts. While ivermectin uptake was encouraging, fear of side effects and reliance on traditional medicine remain barriers to full participation. Addressing these requires community-tailored pharmacovigilance and sustained health education to strengthen trust in ivermectin and promote adherence. Age, education, religion, and place of residence strongly influenced knowledge, attitudes, and practices. The coexistence of onchocerciasis with non-communicable diseases such as arthritis and hypertension further complicates disease control, adding to the community's health burden. Sustained progress toward elimination will require targeted education, stronger pharmacovigilance, increased community engagement, and integrated strategies that address infectious and chronic conditions.

## Limitations

This study has some limitations. The cross-sectional design restricts causal interpretation between knowledge, attitudes, practices, and the occurrence of non-communicable diseases. Data were collected using interviewer-administered questionnaires, which may have introduced recall and social desirability bias, particularly for ivermectin use, reported side effects, and health-seeking behaviors. Additionally, information on non-communicable diseases was self-reported and not clinically verified, which may have led to misclassification. Finally, the study was conducted in a single health district, which may limit the generalizability of the findings to other onchocerciasis endemic settings.

## Supporting information

**S1 File. Study questionnaire.** Questionnaire used to collect socio-demographic, clinical, and behavioral data from participants.
(PDF)

**S1 Data. Dataset underlying the findings.** Anonymized dataset used for statistical analysis in this study.
(XLSX)

## Acknowledgments

We are grateful to the study participants in the Bafut Health District for their cooperation and willingness to share their experiences. We thank the community leaders and health workers, especially the community drug distributors, for their support during the data collection process.

## Author contributions

**Conceptualization:** Irene U. Ajonina-Ekoti, Marcelus U. Ajonina.

**Data curation:** Irene U. Ajonina-Ekoti, Promise A. Aghaeze, Theophilus A. Ekoti, Beri A. Gariba, Moses A. Mbanwi, Adolph A. Fozao, Carine K. Nfor, Marcelus U. Ajonina.

**Formal analysis:** Irene U. Ajonina-Ekoti, Marcelus U. Ajonina.

**Investigation:** Irene U. Ajonina-Ekoti, Tiburce Gangue, Mbunkah D. Achukwi.

**Methodology:** Irene U. Ajonina-Ekoti, Promise A. Aghaeze, Marcelus U. Ajonina.

**Resources:** Irene U. Ajonina-Ekoti.

**Software:** Irene U. Ajonina-Ekoti, Marcelus U. Ajonina.

**Supervision:** Mbunkah D. Achukwi.

**Validation:** Joan Ebanga, Mbunkah D. Achukwi, Marcelus U. Ajonina.

**Visualization:** Irene U. Ajonina-Ekoti.

**Writing – original draft:** Irene U. Ajonina-Ekoti, Theophilus A. Ekoti, Marcelus U. Ajonina.

**Writing – review & editing:** Irene U. Ajonina-Ekoti, Promise A. Aghaeze, Joan Ebanga, Theophilus A. Ekoti, Tiburce Gangue, Beri A. Gariba, Moses A. Mbanwi, Adolph A. Fozao, Carine K. Nfor, Mbunkah D. Achukwi, Marcelus U. Ajonina.

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
