## [Decision Letter · Decision Letter 0]

17 Dec 2025

Response to Reviewers
Revised Manuscript with Track Changes
Manuscript

Shaden Kamhawi

co-Editor-in-Chief

Paul Brindley

co-Editor-in-Chief

**Additional Editor Comments: Journal Requirements:**

1) Please upload all main figures as separate Figure files in .tif or .eps format. For more information about how to convert and format your figure files please see our guidelines: 

2) We have noticed that you have uploaded Supporting Information files, but you have not included a list of legends. Please add a full list of legends for your Supporting Information files after the references list.

3) Some material included in your submission may be copyrighted. According to PLOSu2019s copyright policy, authors who use figures or other material (e.g., graphics, clipart, maps) from another author or copyright holder must demonstrate or obtain permission to publish this material under the Creative Commons Attribution 4.0 International (CC BY 4.0) License used by PLOS journals. Please closely review the details of PLOSu2019s copyright requirements here: PLOS Licenses and Copyright. If you need to request permissions from a copyright holder, you may use PLOS's Copyright Content Permission form.

Potential Copyright Issues:

i) Figure 1. Please (a) provide a direct link to the base layer of the map (i.e., the country or region border shape) and ensure this is also included in the figure legend; and (b) provide a link to the terms of use / license information for the base layer image or shapefile. We cannot publish proprietary or copyrighted maps (e.g. Google Maps, Mapquest) and the terms of use for your map base layer must be compatible with our CC BY 4.0 license.

4) In the online submission form, you indicated that The dataset used in the present study is available from the corresponding author upon reasonable request.. All PLOS journals now require all data underlying the findings described in their manuscript to be freely available to other researchers, either

1. In a public repository

2. Within the manuscript itself

3. Uploaded as supplementary information.

5) Kindly revise your competing statement in the online submission form to align with the journal's style guidelines: 'The authors declare that there are no competing interests.'

**Reviewers' comments:**

**Key Review Criteria Required for Acceptance?**

**Methods**

-Are the objectives of the study clearly articulated with a clear testable hypothesis stated?

-Is the study design appropriate to address the stated objectives?

-Is the population clearly described and appropriate for the hypothesis being tested?

-Is the sample size sufficient to ensure adequate power to address the hypothesis being tested?

-Were correct statistical analysis used to support conclusions?

-Are there concerns about ethical or regulatory requirements being met?

Reviewer #1: The objectives were clearly stated and the study design is appropriate to address the stated objectives.

The population is clearly described but the reason for the choice of only adults in this study was not stated.

The sample size is sufficient and correct statistical analysis were used.

Reviewer #2: (No Response)

**Results**

-Does the analysis presented match the analysis plan?

-Are the results clearly and completely presented?

-Are the figures (Tables, Images) of sufficient quality for clarity?

Reviewer #1: The analysis presented match the analysis plan and results were clearly presented.

Reviewer #2: (No Response)

**Conclusions**

-Are the conclusions supported by the data presented?

-Are the limitations of analysis clearly described?

-Do the authors discuss how these data can be helpful to advance our understanding of the topic under study?

-Is public health relevance addressed?

Reviewer #1: The conclusions are supported by the data presented. The limitations of analysis were clearly described.

The data was well discussed and the public health relevance was addressed.

Reviewer #2: (No Response)

**Editorial and Data Presentation Modifications?**

Reviewer #1: Line numbers should be added

Reviewer #2: (No Response)

**Summary and General Comments**

Reviewer #1: General comments: The study has contributed to KAP regarding onchocerciasis as well as the prevalence of NCDs. However, the authors have not clearly explained how the co-existence of onchocerciasis with NCDs will affect elimination efforts.

Methods

Sample size and sampling method

Why were only adults recruited in the study? What sampling technique was used in selecting the adults in the household?

Results

The results revealed that they had adequate knowledge about onchocerciasis, where did they acquire this knowledge from? Also, how did they learn about filarial worm?

Discussion

Include research that have pointed out cases of ivermectin resistance in Cameroon? Also references on the endemicity of cameroon across all ten regions should be cited.

In your discussion, you stated that comorbid conditions increase clinical burden in endemic areas and complicates case management and elimination efforts. How is this so? Are there documented evidences from health workers regrading this claim?

Why do female gender have positive attitudes compared to males? Are they more infected?

Reviewer #2: Studies as this one help identify the level of awareness and common misconceptions about onchocerciasis, CDTI, and NCDs within the community. Understanding and addressing these misconceptions is essential for effective disease control and prevention. The insights gained from KAP studies can inform the development of policies and programs that are culturally sensitive and community-specific. This ensures that interventions are more likely to be accepted and sustained by the community. Community participation is a key component of the CDTI strategy. By understanding the community's attitudes and practices, health programs can be designed to foster greater community engagement and ownership and CDD/CHW can evaluate the effectiveness of existing health education programs and identify areas that need improvement. Therefore, this is a very important topic that warrants publication.

Major comment: Please add a section on limitation of the study, such as that cross sectional studies as this one do not establish causality. Furthermore, interviewer administered questionaires can introduce bias, self reporting of data, etc.

Minor comments:

Intro:

• Citation needed: NCDs share complications such as vision loss with onchocerciasis, and anecdotal reports suggest that people with conditions like arthritis often avoid ivermectin due to perceived or real side effects.

• Please elaborate on the following link, which is not clear: In Cameroon, where NCDs account for more than 30% of deaths, community health workers report that up to 40% of ivermectin recipients experience adverse effects, further undermining adherence

Methods:

• Data collection instrument: Please elaborate how questions were asked (open, closed etc) and how interviewers were trained to achieve the maximum neutrality.

Discussion:

• Please elaborate further, whether it is the endemicity level or maybe that in hypoendemic areas more rounds of IVM have been distributed with higher coverage and thus awareness is higher due to more active program: “Furthermore, respondents living in hypoendemic areas were more knowledgeable than those in hyper-endemic zones, echoing findings that endemicity level can shape awareness and risk perception. “

• Please a) discuss their relationship with HCW, as other countries have discussed that the need for both may not be necessary and b) unclear whether CDDs have been the interviewers and how they were trained: “Making use of community drug distributors (CDDs), who were the primary source of information in this study, may offer a practical pathway to address these gaps. “

• Please elaborate, as one cause of itching could be the biting of simulia, the other one is the itching induced my mf: “The observation that many symptoms occurred in the morning and afternoon is consistent with the biting activity of Simulium blackflies, which are most active during daylight hours along fast-flowing river basins “

• Please correct, as no causal link can be made in any of the points addressed in this study (albeit the OAE has been found to be associated with onchocerciasis): “While the direct causal links between onchocerciasis and most NCDs remain unclear, recent evidence points to the neurological consequences of chronic infection. “

PLOS authors have the option to publish the peer review history of their article (what does this mean? ). If published, this will include your full peer review and any attached files.

**Do you want your identity to be public for this peer review?** For information about this choice, including consent withdrawal, please see our Privacy Policy .

Reviewer #1: No

Reviewer #2: No

**Figure resubmission:**

**Reproducibility:** To enhance the reproducibility of your results, we recommend that authors of applicable studies deposit laboratory protocols in protocols.io, where a protocol can be assigned its own identifier (DOI) such that it can be cited independently in the future. Additionally, PLOS ONE offers an option to publish peer-reviewed clinical study protocols. Read more information on sharing protocols at https://plos.org/protocols?utm_medium=editorial-email&utm_source=authorletters&utm_campaign=protocols

---

## [Editor Report · Decision Letter 1]

24 Dec 2025

Dear Dr. Ajonina,

We are pleased to inform you that your manuscript 'Onchocerciasis and Non-Communicable Diseases in the Bafut Health District, Cameroon: Knowledge, Attitudes, and Practices Towards Community-Directed Treatment with Ivermectin' has been provisionally accepted for publication in PLOS Neglected Tropical Diseases.

Best regards,

Lucienne Tritten

Section Editor

Shaden Kamhawi

co-Editor-in-Chief

Paul Brindley

co-Editor-in-Chief

Thank you for the prompt revision of your manuscript, which now satisfies all criteria for publication.

---

## [Editor Report · Acceptance letter]

Dear Prof Ajonina,

We are delighted to inform you that your manuscript, "Onchocerciasis and Non-Communicable Diseases in the Bafut Health District, Cameroon: Knowledge, Attitudes, and Practices Towards Community-Directed Treatment with Ivermectin," has been formally accepted for publication in PLOS Neglected Tropical Diseases.

Best regards,

Shaden Kamhawi

co-Editor-in-Chief

Paul Brindley

co-Editor-in-Chief
